# Toward Spec-Driven Code Review Vision: Orchestrating Human–AI Collaboration in Code Review

## Abstract

Code review enforces software standards, architectural constraints, API contracts, and quality expectations in modern development workflows. However, while projects define numerous specifications, current review tools remain largely diff-centric and provide limited support for operationalizing these standards during review. As systems scale and teams become increasingly distributed, enforcing specifications consistently becomes difficult, leading to repeated debates, implicit assumptions, and fragmented knowledge.

The emergence of agentic AI systems introduces new opportunities to structure review more systematically. Rather than merely generating comments or detecting defects, AI agents can assist in identifying relevant specifications, synthesizing validation plans, and executing targeted checks. Inspired by the philosophy of Spec-Driven Development, we argue that code review should be explicitly structured around specifications, in particular specification-driven review planning.

We introduce *Spec-Driven Code Review (SDCR)*, a framework that organizes review into three phases: Review Planning, Review Execution, and Review Artefact Persistence. SDCR treats specifications as first-class review artifacts, enabling human–AI collaboration to produce structured review plans, evidence-linked assessments, and reusable review knowledge. By reframing review as a planning-centric, specification-aware process, SDCR aims to improve scalability, consistency, and accountability in human–AI code review systems.

## Keywords

Code review, Spec-Driven Development

**ACM Reference Format:**

Anonymous Author(s). 2024. Toward Spec-Driven Code Review Vision: Orchestrating Human–AI Collaboration in Code Review. In *Proceedings of Spec-Driven Development (AIware 2026)*. ACM, New York, NY, USA, 5 pages. https://doi.org/XXXXXXX.XXXXXXX

## 1 Introduction

Code review is a foundational practice in modern software engineering [1, 2]. It enforces coding standards, preserves architectural integrity, safeguards API contracts, and ensures adherence to security, performance, and maintainability guidelines [3]. Mature projects define explicit and implicit standards that govern how changes should be evaluated. In principle, review serves as the

mechanism through which these standards are enforced and negotiated over time. Formal process standards, such as IEEE 1028 for software reviews and audits [4], as well as lifecycle models including ISO/IEC/IEEE 12207 [5], define structured review practices and roles.

However, modern code review platforms provide limited support for systematically operationalizing structured review standards within everyday workflows [6, 7]. While these platforms effectively facilitate discussion and change inspection, they typically do not make relevant specifications explicit nor provide mechanisms to formally link review activities to predefined standards. As a result, reviewers must reconstruct applicable constraints from documentation, prior decisions, and institutional memory [8–11]. This reconstruction process introduces ambiguity and increases the likelihood of inconsistent interpretations of the same change [12–15]. Prior studies have shown that the quality and rigor of code review directly influence overall software quality [16–18]. When standards are not systematically enforced or clearly articulated during review, variability in review practices can propagate to variability in code quality. Introducing greater structure into the planning and execution of reviews is therefore essential for improving review consistency and, ultimately, enhancing software quality.

The emergence of agentic AI systems introduces both new challenges and new opportunities [19]. AI agents are increasingly capable of assisting in review tasks, from generating comments to executing analyses and synthesizing multi-step plans. Yet without explicit representations of the standards governing a change, AI systems risk operating at a superficial level, focusing on localized issues rather than principled validation. At the same time, agentic AI frameworks demonstrate the ability to reason over artifacts, construct structured plans, and coordinate tool execution, suggesting that development workflows can become more structured and goal-oriented. In parallel, specification-centric development practices are gaining renewed attention in industry; for example, initiatives such as GitHub's Spec-Kit[1] aim to represent specifications explicitly within repositories to better support automation and tooling. These developments indicate a broader shift toward treating specifications not merely as documentation, but as operational artifacts that guide development processes. Inspired by this convergence of agentic AI capabilities and specification-driven tooling, we argue that code review should likewise be structured around explicit specifications, enabling human reviewers and AI systems to engage in principled, specification-driven review planning rather than ad hoc inspection.

In this paper, we introduce **Spec-Driven Code Review (SDCR)**, a framework that reorients review around three phases: Review Planning, Review Execution, and Review Artifact Persistence. SDCR first identifies the specifications relevant to a given change, translates them into a structured review plan, executes targeted checks to gather evidence, and produces a spec-by-spec assessment. All

---

[1]https://github.com/github/spec-kit

review artifacts are then archived to enable reuse and long-term consistency. By making specifications explicit and integrating human judgment with agentic execution loops, SDCR aims to improve scalability, consistency, and transparency in modern code review.

We present the conceptual foundations of SDCR, detail its workflow and artifacts, and discuss the research challenges required to realize this vision in practice. As agentic AI becomes increasingly embedded in development workflows, structuring review around explicit specifications is essential for aligning human intent, project standards, and automated reasoning.

## 2 Limitations of Current Code Review Practices

Despite its central role, contemporary code review still offers limited support for *systematically operationalizing* review standards inside the day-to-day workflow. In practice, review tools make it easy to inspect changes and exchange comments, but they rarely help reviewers (or tools) (i) surface the *relevant constraints/specifications* that should govern the change, (ii) translate those constraints into an explicit, shareable *review plan*, and (iii) connect review actions (checks, tests, evidence) back to those constraints. As a result, reviewers repeatedly "reconstruct" what should be validated from scattered documentation, past discussions, and personal experience [11]. This reconstruction is a known source of friction in modern code review: practitioners report challenges such as obtaining timely feedback and handling large/complex reviews, both of which degrade review throughput and consistency [? ]. More importantly, reconstruction creates ambiguity: when rationale or non-functional expectations are not made explicit, reviewers can become confused about what the change is *supposed* to satisfy, which increases the likelihood of confusion in reviewers' comments [13] and divergent interpretations and prolonged back-and-forth, as demonstrated by Hiro et al. [9] where up to 37% of code changes have divergent reviewers scores. Empirical evidence also suggests that review outcomes are tightly coupled with software quality outcomes: code review is expected to improve quality, and studies show links between review activity and quality-related phenomena (e.g., changes in design issues), yet improvements may occur indirectly and inconsistently—precisely because validation is not always guided by explicit quality constraints [? ]. In other words, when standards remain implicit, "good review" depends on who the reviewers are, what they remember, and how much time they have—making enforcement variable and repeatedly negotiated rather than consistently operationalized [15, 16].

These limitations become more pronounced in the context of emerging agentic software engineering workflows. As AI systems increasingly accelerate code generation and iteration, the volume and velocity of submitted changes are expected to grow, heightening the risk that code review becomes a structural bottleneck rather than a scalable quality gate [19]. Industry efforts already treat review latency and throughput as first-class scaling challenges, proposing interventions—such as behavioral nudges—to reduce review delays at scale [? ]. Such work implicitly underscores that maintaining review efficiency is difficult even before accounting for AI-amplified change production.

Despite advances in AI-assisted review tools, most systems emphasize localized assistance—such as drafting comments, suggesting fixes, or flagging defects—rather than addressing the broader planning problem of review. Specifically, they do not systematically determine which specifications (e.g., API contracts, architectural constraints, security requirements, performance budgets, or release policies) are relevant for a given change, nor do they organize validation activities into a coherent, auditable review plan. Prior work further highlights the importance of structured guidance: Gonçalves et al. [10] show that providing reviewers with guided checklists improves review effectiveness and efficiency, reinforcing the value of explicit validation structure.

Without specification-aware planning, both human reviewers and AI agents remain confined to reactive coordination patterns—responding to diffs and comments, negotiating which expectations apply, and re-litigating similar issues across changes. Such unstructured workflows increase ambiguity and contribute to repeated debates and unclear rationale, phenomena previously associated with confusion in review processes [13]. These observations motivate SDCR's central claim: scaling code review in an agentic era requires moving beyond ad hoc inspection toward *specification-anchored review planning*, where validation activities are explicitly selected, executed, and recorded with traceable links to governing specifications, thereby enabling clearer coordination, auditability, and reuse across future reviews.

## 3 SDCR overview

Figure 1 presents the SDCR framework, structured into three phases: *Review Planning*, *Review Execution*, and *Review Artifact Persistence*. SDCR treats review as a specification-oriented, human–AI collaborative process in which specifications guide planning, execution produces evidence, and structured artifacts are persisted to support future reuse.

### 3.1 Phase I: Review Planning

The Review Planning phase establishes what should be checked and why before any validation is executed. Rather than starting from a line-by-line diff inspection, SDCR first makes relevant specifications explicit and translates them into a structured plan.

*3.1.1 Step 1: Identify Relevant Specifications.* Given a newly submitted code change, AI agents identify specifications that may be affected. These specifications can originate from project policies, API contracts, architectural constraints, performance budgets, security rules, test invariants, or prior review decisions. The output of this step is a *selected specification set* that makes explicit the expectations governing the change. By surfacing these specifications early, SDCR reduces hidden assumptions and aligns reviewers around a shared validation scope.

*3.1.2 Step 2: Translate Specifications into a Review Plan.* For each identified specification, the AI agent translates abstract requirements into a concrete review plan. This plan specifies which artifacts must be examined, which checks should be executed, what evidence must be collected, and what criteria determine satisfaction. The resulting *review plan* operationalizes specifications into actionable validation steps, transforming review from informal inspection into structured evaluation.

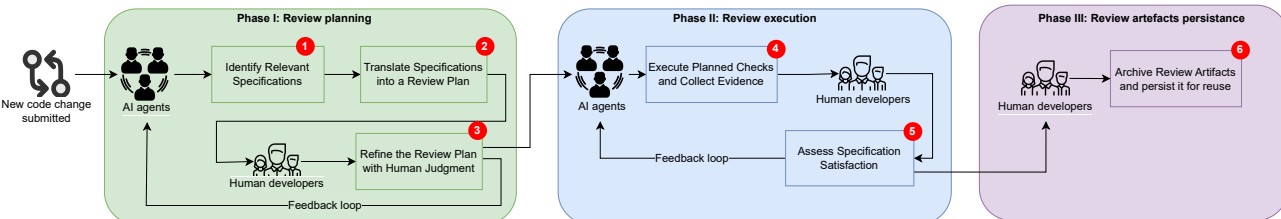

**Figure 1: SDCR overview.**

*3.1.3  Step 3: Refine the Review Plan with Human Judgment.* Human developers then review and refine the proposed review plan. They may add missing specifications, remove irrelevant checks, adjust thresholds, or clarify ambiguous requirements. This feedback loop ensures that the plan reflects contextual knowledge and project-specific constraints. The result of this step is a *validated review plan* that balances automated reasoning with human expertise.

## 3.2  Phase II: Review Execution

The Review Execution phase validates the refined plan and produces evidence grounded in actual system behavior and analysis results.

*3.2.1  Step 4: Execute Planned Checks and Collect Evidence.* AI agents execute the validated review plan by running the specified tests, analyses, and inspection procedures. This may involve invoking test suites, static analysis tools, performance benchmarks, or custom scripts. The outcome of this step is an *evidence bundle* explicitly linked to each specification. This shifts review discussion from opinion-based arguments to evidence-based assessment.

*3.2.2  Step 5: Assess Specification Satisfaction.* Using the collected evidence, the AI agent produces a structured assessment of each specification, indicating whether it is satisfied, violated, or inconclusive. Human developers review this assessment, provide interpretation where necessary, and may trigger additional execution cycles through the feedback loop. This step converges toward a *structured review outcome* grounded in specification-level reasoning rather than diff-level commentary.

## 3.3  Phase III: Review Artifact Persistence

The final phase ensures that review knowledge is not lost once a decision is made.

*3.3.1  Step 6: Archive Review Artifacts for Reuse.* SDCR archives the complete set of review artifacts, including the selected specifications, the refined review plan, the execution trace, the collected evidence, and the final structured assessment. This persisted record forms a *spec-anchored review memory* that can be reused in future reviews of similar changes. When specifications are refined or reinterpreted during review, those updates are captured explicitly, enabling review knowledge to accumulate over time rather than being buried in ephemeral comment threads.

## 3.4  SDCR in practice: An illustrative example

To illustrate the SDCR framework, consider a backend service where a developer submits a change that introduces caching into a public

API method to improve response time. The modification alters internal logic and introduces a shared in-memory cache.

**Phase I: Review Planning.** Upon submission of the change, the AI agent identifies relevant specifications. Based on the modified files and API boundaries, it selects the following specifications: (i) public API backward compatibility must be preserved, (ii) performance regressions beyond a predefined threshold are not permitted, and (iii) shared state must not introduce concurrency defects. These form the *selected specification set.*

The agent then translates these specifications into a review plan. For backward compatibility, it proposes running an API-diff tool and regression tests. For performance, it recommends executing benchmark suites and comparing latency metrics against historical baselines. For concurrency safety, it suggests running thread-safety tests and static analysis checks for shared-state access. This results in a structured review plan that links each specification to required artifacts, checks, and acceptance criteria.

Human reviewers inspect the plan and refine it by adding a stress-test scenario previously used for similar caching changes. They also clarify that performance improvements must not increase memory usage beyond a specified threshold. The review plan is validated and finalized.

**Phase II: Review Execution.** The AI agent executes the validated plan. It runs the API-diff tool, confirming that method signatures remain unchanged. Regression tests pass. Benchmark results indicate a 15% latency improvement but reveal increased memory consumption. Concurrency analysis detects a potential race condition under high load. These outputs are collected into an *evidence bundle*, explicitly linked to each specification.

The agent produces a structured assessment: backward compatibility is satisfied, performance goals are partially satisfied but memory thresholds are exceeded, and concurrency safety is inconclusive due to the detected race condition. Human reviewers examine the evidence and request an additional synchronization mechanism. The developer updates the code, triggering another execution cycle.

**Phase III: Review Artifact Persistence.** After the concurrency issue is resolved and benchmarks meet defined thresholds, the review is finalized. The structured outcome records specification-level judgments and associated evidence. SDCR archives the selected specifications, the refined review plan, execution traces, benchmark reports, and the final assessment as a reusable review artifact.

In future reviews involving caching modifications, the system can automatically retrieve the relevant specifications and reuse

the validated review plan template, reducing repeated debate and improving consistency across similar changes.

## 4 Implications

The Spec-Driven Code Review (SDCR) vision has implications for both research and practice. By treating specifications as first-class review artifacts and centering review planning as a structured human–AI activity, SDCR opens several new directions for empirical study, system design, and organizational adoption.

### 4.1 Implications for Researchers

**Rethinking Review Planning as a Research Problem.** Current research on code review predominantly focuses on defect detection, reviewer recommendation, comment generation, and prediction tasks. SDCR suggests that *review planning itself* is an underexplored research dimension. Future work can investigate how to automatically identify relevant specifications for a given change, how to formalize specification-to-plan translation, and how to evaluate the quality of generated review plans.

**Specification Mining and Representation.** A core requirement of SDCR is the availability of machine-interpretable specifications. This raises research questions on mining specifications from documentation, prior reviews, architectural artifacts, and project histories. Researchers must explore how to represent specifications in forms that support reasoning, traceability, and evolution over time.

**Human–AI Co-Planning Interfaces.** SDCR emphasizes iterative refinement between AI agents and human reviewers during the planning phase. This calls for new interaction models and interface designs that enable transparent plan generation, critique, and revision. Studying how developers trust, modify, or override AI-generated review plans becomes central to understanding effective collaboration.

**Evaluation Beyond Comment Quality.** Traditional evaluation metrics for AI-assisted review focus on comment usefulness or defect detection accuracy. SDCR shifts attention toward plan completeness, specification coverage, decision consistency, and long-term knowledge reuse. Researchers must define new benchmarks and datasets to measure planning effectiveness and specification alignment.

### 4.2 Implications for Practitioners

**Making Standards Operational Rather Than Implicit.** SDCR encourages teams to make their standards and expectations explicit and link them directly to review activities. This can reduce repeated debates and inconsistent interpretations, particularly in large or distributed teams.

**Improving Review Scalability.** As systems grow, it becomes difficult to ensure consistent enforcement of architectural and quality constraints. Spec-driven planning allows organizations to systematically associate checks and evidence with predefined expectations, improving scalability without relying solely on individual reviewer expertise.

**Supporting Onboarding and Knowledge Retention.** By archiving structured review artifacts, SDCR transforms review decisions into reusable knowledge. New team members can understand not only what decisions were made, but which specifications governed those decisions and how evidence was collected.

**Enabling Accountable AI in Review Pipelines.** For teams integrating agentic AI into review workflows, SDCR provides a principled structure for oversight. AI agents operate within explicitly defined specifications and review plans, reducing the risk of opaque or misaligned automated judgments.

Overall, SDCR reframes code review from an informal diff-centric practice into a specification-aware, planning-driven human–AI process, with significant implications for how review systems are designed, evaluated, and adopted.

## 5 Conclusion and Future Directions

Code review remains a central mechanism for enforcing software standards, yet existing review workflows largely operate in a diff-centric manner, leaving specifications implicit and inconsistently applied. As systems scale and AI agents become increasingly integrated into development pipelines, the need for structured, specification-aware review processes becomes more pressing.

In this paper, we introduced *Spec-Driven Code Review (SDCR)*, a vision that reorients review around explicit specifications and, critically, around structured review planning. By treating specifications as first-class review artifacts, SDCR organizes review into three phases: Review Planning, Review Execution, and Review Artifact Persistence. This planning-centric perspective enables human–AI collaboration to move from reactive inspection to principled validation grounded in explicit expectations. We argue that such structuring is essential for achieving scalable, consistent, and accountable code review in agentic environments.

This vision opens several avenues for future work. First, research is needed on automatically identifying and representing relevant specifications for a given change, including mining specifications from documentation, architectural artifacts, and historical reviews. Second, formal methods for translating specifications into executable review plans must be developed and evaluated. Third, new interaction models are required to support iterative human–AI co-planning and transparent refinement of review strategies. Fourth, empirical studies should investigate how specification-driven planning affects review consistency, disagreement rates, onboarding efficiency, and long-term knowledge retention.

Beyond technical challenges, future work should also examine organizational adoption: how teams externalize implicit standards, how specification repositories evolve, and how spec-driven review integrates with continuous integration and agentic development pipelines. Ultimately, advancing SDCR requires interdisciplinary efforts spanning software engineering, human–AI interaction, and specification engineering.

By reframing code review as a planning-centric, specification-driven human–AI process, this work aims to stimulate a new research direction toward more principled and scalable review systems.

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

Received 20 February 2024; revised 12 March 2024; accepted 5 June 2024

