# OpenReview forum: "Toward Spec-Driven Code Review Vision: Orchestrating Human–AI Collaboration in Code Review"
_ACM.org/AIWare/2026/Conference — Submitted to AIware 2026_

### Official Review · Reviewer_tkFd · 2026-03-07

**Rating:** 2
**Confidence:** 5

**Review:**

Strengths:

The study is relevant to the Software Engineering community.
Standards are employed in many areas to improve and stabilize processes, therefore they are welcome in the code review process.
Well-organized repository with extensive documentation.

Weaknesses:

No comparison with similar initiatives was presented.
Lack of evaluation.

The authors provided a repository with the implementation of the tool.

The text is easy to read; However, it consistently presents problems in proving the lack of review:

Without specification-driven planning, both human reviewers and AI agents remain confined to reactive coordination patterns — responding → lack of alignment with the right margin of the page.

Missing references:
both degrade review productivity and consistency [?].

is not always guided by explicit quality constraints [?].

to reduce delays in large-scale review [?].

The study proposed a framework for integrating standards into code review, making it less susceptible to variation among programmers in terms of experience, style, etc.

It is unclear how the study positions itself in relation to previous studies; in fact, this limits its novelty and relevance.

Alami, Adam, Victor Jensen, and Neil Ernst. "Accountability in code review: The role of intrinsic drivers and the impact of LLMs." ACM Transactions on Software Engineering and Methodology 34.8 (2025): 1-44.

Cihan, Umut, et al. "Automated code review in practice." 2025 IEEE/ACM 47th International Conference on Software Engineering: Software Engineering in Practice (ICSE-SEIP). IEEE, 2025.

The authors did not present a metric for comparison with similar proposals, nor did they conduct experiments to empirically evaluate the approach, which affected its rigor. Details about LLM training, prompt strategies, version, and parameters are not in the text nor discussed.

**Summary:**

This article presents SDCR - Specification-Driven Code Review: an organizational framework for handling code reviews composed of 3 stages: planning, execution, and persistence, in collaboration with humans to improve scalability. The study presents a possible use case.

---

> ### Author Response · Authors · 2026-03-18
>
> **C3.1/C3.6: No comparison with similar initiatives was presented. Lack of evaluation.
> The authors provided a repository with the implementation of the tool.**
>
> **Response:**  We thank the reviewer for this comment and for pointing to relevant recent work in C3.6. We clarify that SDCR is positioned as a complementary direction to existing research on automated code review and AI-assisted development. Prior work such as Alami et al. (2025) investigates the social and behavioral aspects of code review, including accountability and how it is affected by LLM-assisted review. Cihan et al. (2025) studies the practical impact of LLM-based automated code review tools in industry, focusing on aspects such as defect detection, comment usefulness, and developer perception. More broadly, existing work either focuses on generating review feedback (e.g., comments, defect detection) or evaluating correctness against predefined validation rules. In contrast, SDCR introduces a specification-driven planning layer that determines which constraints are relevant for a given change and organizes validation activities accordingly.
>
> **Action:** We will revise Section 2 to explicitly position SDCR relative to prior work, including recent studies on AI-assisted code review and automated review practices, and clarify how SDCR complements these approaches by introducing specification-driven review planning.
>
>
> **C3.2: The text is easy to read; However, it consistently presents problems in proving the lack of review:
> Without specification-driven planning, both human reviewers and AI agents remain confined to reactive coordination patterns — responding → lack of alignment with the right margin of the page.
> Missing references: both degrade review productivity and consistency [?]. is not always guided by explicit quality constraints [?]. to reduce delays in large-scale review [?].**
>
> **Response:** We thank the reviewer for identifying these issues (missing references and overflow).
>
> **Action:** Revise the affected sentences for clarity and add the missing references.
>
> **C3.5: The study proposed a framework for integrating standards into code review, making it less susceptible to variation among programmers in terms of experience, style, etc.**
>
> **Response:** We appreciate this interpretation. One of SDCR’s goals is precisely to make review standards explicit and systematically integrated into review workflows.
>
> **Action:** Clarify this motivation in the Introduction.
>
> **C3.6: It is unclear how the study positions itself in relation to previous studies; in fact, this limits its novelty and relevance.**
>
> Alami, Adam, Victor Jensen, and Neil Ernst. "Accountability in code review: The role of intrinsic drivers and the impact of LLMs." ACM Transactions on Software Engineering and Methodology 34.8 (2025): 1-44.
>
> Cihan, Umut, et al. "Automated code review in practice." 2025 IEEE/ACM 47th International Conference on Software Engineering: Software Engineering in Practice (ICSE-SEIP). IEEE, 2025.
>
> **Action:** As explained in the answer of C3.1, we will revise Section 2 to explicitly position SDCR relative to prior work, including recent studies on AI-assisted code review and automated review practices, and clarify how SDCR complements these approaches by introducing specification-driven review planning.
>
>
> **C3.7: The authors did not present a metric for comparison with similar proposals, nor did they conduct experiments to empirically evaluate the approach, which affected its rigor. Details about LLM training, prompt strategies, version, and parameters are not in the text nor discussed.**
>
> **Response:** We thank the reviewer for this important point. We acknowledge that the paper does not include empirical evaluation or LLM-specific implementation details, as SDCR is proposed as a vision framework rather than a realized system.
>
> Our goal is to introduce specification-driven review planning as a new research direction. However, we agree that evaluation and system realization are essential. Building on existing automated code review practices, which typically evaluated using metrics such as defect detection effectiveness, usefulness of feedback, and developer adoption; SDCR would additionally require metrics such as specification coverage, plan completeness, traceability between specifications and evidence, and review consistency.
>
> Regarding LLM-specific details (e.g., prompts, models), SDCR does not assume a particular configuration; these are implementation choices. The framework instead defines the structure within which such models operate.
>
> **Action:** We will revise the Introduction and Section 4 to clearly position SDCR as a vision framework and explicitly outline evaluation strategies, metrics, and implementation considerations.

---

### Official Review · Reviewer_4rtY · 2026-03-08

**Rating:** 3
**Confidence:** 4

**Review:**

strength:
-  The paper proposes a framework that aims to improve the structure and consistency of the code review process.

negative:
- Introduces several hand-wavy claims and does not clearly define what constitutes a specification
- The proposed agentic workflow is very general. Agent compliance is not discussed.

comments:
- Page 2, line 144: fix the citation placeholder `[?]`.
- Page 2, line 156: fix the citation placeholder `[?]`.
- Page 2, line 169: fix the citation placeholder `[?]`.
- Page 2, line 186: formatting issue with overflowing text.
- The paper claims that code review is largely ad-hoc, but many organizations employ structured review guidelines. The authors should clarify the empirical basis for this claim.
- The paper treats specifications broadly, including formal specifications, policies, test cases, architectural guidelines, and historical review decisions. These artifacts differ in structure and formality. As a result, it is unclear:
    - what exactly qualifies as a specification,
    - where these specifications originate,
    - how they are represented in a machine-interpretable form
    - how the proposed AI agent would identify them automatically
    - In many real-world software projects, documentation is incomplete or informal. This raises concerns about the feasibility of automatically identifying and operationalizing specifications, and the system will suffer from a “garbage in, garbage out” problem.
- The framework assumes that an AI agent can identify relevant specifications and translate them into a structured review plan. However, the paper does not describe how these tasks would be performed. These steps are the core technical challenges of the proposed approach. In particular, the paper does not discuss:
    - how specifications are detected or retrieved,
    - how they are translated into executable review plans, and
    - how specification derived from agent would be evaluated.
    Thus, the framework remains largely conceptual and hand-wavy.
- The example in Section 3.4 also appears similar to a typical CI/CD workflow where tests, benchmarks, and static analysis tools are executed to validate changes. The paper would benefit from clarifying how SDCR offers new contribution.
- Paper should mention what is end to end process in code review and actually specify possible bottleneck. For example, the paper discusses reconstruction, but what does this mean?
- The paper introduces a lot of terms, yet provides little support how to measure accuracy of divergence, correct reconstruction, as well as incorporate non-functional properties?
- Even if such a pipeline could be implemented, it is unclear whether developers would trust or adopt the system in practice. Code review often involves contextual judgment among developers, and it is not clear how well a specification-driven pipeline would integrate with existing workflows.
- The paper sometimes gives the impression that AI agents are assumed to solve many problem without discussion challenges involved. The paper may benefit from discussing limitations and technical difficulties of applying agent-based approaches to code review.

**Summary:**

This paper presents a vision for Spec-Driven Code Review (SDCR), which structures code review into three phases: review planning, review execution, and review artifact persistence. While the idea of organizing code review around specifications is interesting, the paper remains highly conceptual and leaves several key aspects unclear.

---

> ### Author Response · Authors · 2026-03-18
>
> **C2.1 -- C2.4: Missing citations and formatting**
>
> **Response:** We thank the reviewer for identifying the overflowing text.
>
> **Action:** We correct this issue in the paper.
>
> **C2.5: Code review not always ad-hoc**
>
> **Response:** We agree that many organizations use structured review guidelines. As cited, standards such as ISO/IEC/IEEE 12207 and IEEE 1028 recommend guideline- and checklist-based reviews, especially in safety-critical domains. Our goal is to generalize this good practice across all domains. Current platforms (e.g., GitHub, Gerrit) mainly support diff-based review and offer limited support for operationalizing such guidelines. SDCR aims to extend this by surfacing relevant specifications and structuring review activities around them.
>
> **Action:** Clarify that SDCR generalizes specification- and guideline-driven review beyond safety-critical domains and integrates it into modern code review tools.
>
> **C2.6: Definition of specifications unclear**
>
> **Response:**We agree that specifications vary in structure and formality. In SDCR, we use the term broadly to refer to constraints governing system behavior (e.g., API contracts, architectural rules, or policies) from sources such as documentation, CI, tests, or review history. These artifacts can be surfaced using artifact-specific retrieval mechanisms. For example, ecosystems like OpenStack maintain repository-based specifications (e.g., Nova specs), showing that structured specifications already exist in practice. SDCR does not assume fully formal specifications; instead, AI surfaces candidate constraints from heterogeneous artifacts, while human reviewers validate and refine them before generating review plans.
> **Action:** Add this clarification to Section 3.1.1
>
>
> **C2.7: Core tasks under-specified**
>
> **Response:**SDCR can be viewed as a pipeline with three stages: (1) specification identification, (2) review plan generation, and (3) execution and assessment. Given a code change, the AI analyzes the change scope (e.g., modified files, intent) using techniques like dependency and impact analysis, retrieves relevant specifications from heterogeneous sources, and maps them to validation actions to form a structured review plan.
> Human reviewers refine the plan before execution. The AI executes it, collects evidence, and produces structured outcomes (e.g., satisfied / violated), which are interpreted by humans. Each result is linked to its originating specification, ensuring traceability with human oversight.
>
> **Action:** We will incorporate this clarification into Section 3 and extend Section 4 to explicitly discuss the associated challenges (e.g., specification retrieval, boundary identification, and specification-to-plan translation) as key research directions.
>
> **C2.8: Similarity to CI/CD pipelines**
>
> **Response:** SDCR is not equivalent to CI/CD workflows. CI pipelines execute predefined checks uniformly across changes, while SDCR introduces a planning layer that selects and organizes validation activities based on relevant specifications. In this sense, CI focuses on executing checks, whereas SDCR focuses on deciding what to execute and why, enabling more targeted and context-aware validation.
>
> **Action:** Clarify in Section 3.4 the distinction between CI pipelines and SDCR’s specification-driven planning approach.
>
> **C2.9:End-to-end process unclear**
>
> **Response:** “Reconstruction” refers to the process by which reviewers infer relevant constraints by consulting documentation, prior discussions, or experience when these constraints are not explicitly surfaced.
>
> **Action:** Clarify the code review process, define reconstruction, and provide a clearer end-to-end workflow description.
>
> **C2.10: Evaluation metrics unclear**
>
> **Response:** We agree that defining appropriate evaluation metrics for specification-driven review systems remains a challenge. While this paper introduces the SDCR vision, several possible evaluation directions include specification coverage, plan completeness, review consistency, and traceability between specifications, checks, and evidence.
>
> Action: Acknowledge that evaluation is an open research challenge and outline these possible evaluation directions in Section 4.
>
> **C2.11: Adoption concerns**
>
> **Response:** SDCR is designed to support human reviewers rather than replace them. Human reviewers remain responsible for refining plans and making final decisions.
>
> **Action:** Clarify that SDCR is a human-centred framework.
>
> **C2.12: AI agents appear too powerful**
>
> **Response:** We agree that applying AI agents introduces challenges. In SDCR, AI assists by identifying specifications and proposing review plans rather than acting autonomously. The system remains governed by human reviewers, who can refine or override the plan, ensuring human judgment stays central.
>
> **Action:** Clarify SDCR’s human-in-the-loop design, where AI assists in structuring review plans while humans validate, refine, and control the workflow.

---

### Official Review · Reviewer_RXhK · 2026-03-11

**Rating:** 2
**Confidence:** 5

**Review:**

Strengths:
----------------------------
+ Addresses a timely and important problem: how code review should evolve as AI-generated and AI-assisted development increases review volume and complexity.
+ Central motivation is easy to understand and potentially valuable: making specifications explicit could reduce ambiguity, repeated debate, and reviewer inconsistency.
+ Clear three-phase structure: the SDCR framework is logically organized and easy to follow.
+ Actionable research agenda in Section 4

Weaknesses:
------------------------------
- The novelty of SDCR as a new framework is unclear. Specification-based testing, model-driven verification, and formal method-based review approaches have long existed in the literature. What is the limitation of such approaches that could be tackled by the SDCR?
- The paper uses "specification" broadly to encompass API contracts, architectural constraints, performance budgets, security rules, test invariants, and prior review decisions. These artifact types have fundamentally different representations, accessibility, and formalization levels. Section 3.1.1 mentions AI gent identify specification for a newly submitted code change. However, did not mention how AI agents would uniformly identify, parse, or reason over such heterogeneous sources.
- The paper presents a very high-level plan with no pilot study, no prototype implementation, no user survey, and no empirical motivating data. It is expected to include at least some preliminary evidence (e.g., a small-scale feasibility study, a corpus analysis of how specifications appear in real repositories, or an expert survey) that justifies the proposed direction of its own.
- The paper frequently implies that SDCR can improve scalability, consistency, transparency, accountability, onboarding, and knowledge retention. These are attractive outcomes, but the paper currently presents them more as expected benefits than as hypotheses with clearly stated assumptions and constraints.
- The paper repeatedly invokes "human–AI collaboration" but offers minimal concrete description. Who initiates what? How does the human reviewer provide refinement feedback to the AI in Phase I Step 3 through natural language, UI controls, or structured edits? What happens when the human disagrees with the AI-generated review plan? The feedback loops in Figure 1 are confusing. Does the AI agent re-identify the specification again after each conflict?
- Phase III (Review Artifact Persistence) is described in only a few sentences and the illustrative example. There is no discussion of the actual storage format, retrieval mechanism, similarity matching for "future reviews of similar changes," or the lifecycle management of archived specifications.
- Generating a structured review plan per pull request, running multiple AI-driven analysis loops, and archiving full evidence bundles for every change could significantly increase review latency and infrastructure cost. How does SDCR tackle that?
- Missing citation — "?" Placeholders Remain in the Text (noticed in three places)

**Summary:**

This paper introduces Spec-Driven Code Review (SDCR), a vision framework that proposes reorienting code review around explicit specifications rather than the current diff-centric paradigm. The framework organizes review into three phases: Review Planning, Review Execution, and Review Artifact Persistence and positions AI agents as collaborators in identifying relevant specifications, translating them into structured review plans, executing checks, and persisting review knowledge for future reuse.

---

> ### Author Response · Authors · 2026-03-18
>
> **C1.1: Novelty unclear vs specification-based testing / formal methods.**
>
> **Response:**SDCR addresses a complementary but underexplored problem compared to specification-based testing and formal methods, which assume known specifications and fixed validation steps. In contrast, modern AIWare makes it difficult to identify which specifications are relevant for each change. SDCR introduces a planning layer that selects and organizes specifications to build a tailored review plan. While existing approaches verify correctness, SDCR structures how relevant correctness criteria are identified and applied during review.
>
> **Action:** Revise the Introduction and Section 2 to clearly position SDCR relative to specification-based testing, formal methods, and CI pipelines, emphasizing its focus on specification-driven review planning.
>
> **C1.2: Specifications are heterogeneous; unclear how AI identifies them.**
>
> **Response:** We agree that specifications differ in structure and representation. SDCR does not assume a uniform representation; instead, specifications originate from heterogeneous sources such as documentation, CI configurations, tests, architectural descriptions, and review history. These mechanisms enable the system to reason over heterogeneous specifications by mapping them into a common intermediate representation (e.g., normalized constraints or validation intents that can be directly mapped to validation actions) used for review planning.
>
> **Action:** Clarify Section 3.1.1 by explicitly describing different specification sources and how they can be surfaced using artifact-specific retrieval mechanisms (e.g., documentation retrieval, test metadata extraction, and CI configuration parsing).
>
> **C1.3: No pilot study/prototype.**
>
> **Response:** This paper is a vision paper that introduces a research direction rather than a fully implemented system. Its goal is to define the conceptual framework and outline key challenges for specification-driven review. As such, it focuses on motivating the direction and identifying future research rather than providing empirical validation.
> **Action:** Explicitly position the paper as a vision and research agenda in the Introduction and Conclusion, clarifying that empirical validation is future work.
>
>
> **C1.4: Benefits stated as claims rather than hypotheses.**
>
> **Response:** We agree that these statements should be framed as expected outcomes rather than established results.
>
> **Action:** Revise the text to present these benefits as hypotheses and research motivations, clarifying assumptions and conditions under which they may hold.
>
> **C1.5: Human–AI collaboration unclear.**
>
> **Response:** In SDCR, the human can initiate the AI to perform the review, and the AI agent proposes an initial review plan, which human reviewers then refine by adding or modifying review plans and checks. In case of disagreement, reviewers can modify or override the proposed review plan, and optionally trigger re-generation of the plan by the AI, enabling iterative refinement under human control. The AI then executes the validated plan and presents evidence for human interpretation.
>
> **Action:** Clarify the human–AI interaction protocol in Section 3 and revise Figure 1 to make the feedback loop and responsibilities more explicit.
>
> **C1.6: Artifact persistence under-specified.**
>
> **Response:** These artifacts can be stored as repository-linked metadata (attached to pull requests), in external knowledge bases indexed by components or specification types, or in machine-readable formats such as JSON/YAML. Reuse can be supported through similarity-based retrieval (e.g., using embeddings of code changes or specifications).
>
> **Action:** Expand Section 3.3 to briefly describe possible storage representations and retrieval mechanisms for archived review artifacts.
>
> **C1.7: Cost and latency concerns.**
>
> **Response:** SDCR does not assume that all validation checks are executed for every change. Instead, checks are selected based on the relevant specifications identified during the planning phase. For example, documentation updates or other minor changes may only trigger lightweight checks, while more complex modifications may require deeper analysis. In addition, the AI agent only assists in structuring the review plan, while human reviewers remain in control of refining, approving, or modifying the plan before execution.
>
> **Action:** Clarify in Section 3 that checks are selectively triggered based on the nature of the change (e.g., lightweight for minor updates), and that AI assists in structuring review plans while humans retain control over final decisions, limiting overhead.
>
> **C1.8: Missing citation**
>
> **Response:** We thank the reviewer for identifying the missing citation placeholders.
>
> **Action:** Replace the placeholders with the appropriate references in the revised manuscript.

---

### Author Response · Authors · 2026-03-18

We sincerely thank all reviewers for their insightful and constructive feedback. We appreciate the positive assessment of the clarity and motivation of the paper, as well as the detailed comments that helped us identify areas for improvement. In the revision, we will focus on clarifying the positioning and novelty of SDCR, better defining the notion of specifications and their sources, and making the human–AI interaction and system workflow more explicit. We will also revise the manuscript to clearly present SDCR as a vision and research agenda, and address all formatting and missing citation issues. We believe these changes will significantly strengthen the paper and better communicate its contributions.